# LANGUAGE MODELS AS SEMANTIC INDEXERS

## ABSTRACT

Semantic identifier (ID) is an important concept in information retrieval that aims to preserve the semantics of objects such as documents and items inside their IDs. Previous studies typically adopt a two-stage pipeline to learn semantic IDs by first procuring embeddings using off-the-shelf text encoders and then deriving IDs based on the embeddings. However, each step introduces potential information loss, and there is usually an inherent mismatch between the distribution of embeddings within the latent space produced by text encoders and the anticipated distribution required for semantic indexing. Nevertheless, it is non-trivial to design a method that can learn the document's semantic representations and its hierarchical structure simultaneously, given that semantic IDs are discrete and sequentially structured, and the semantic supervision is deficient. In this paper, we introduce LMINDEXER, a self-supervised framework to learn semantic IDs with a generative language model. We tackle the challenge of sequential discrete ID by introducing a semantic indexer capable of generating neural sequential discrete representations with progressive training and contrastive learning. In response to the semantic supervision deficiency, we propose to train the model with a self-supervised document reconstruction objective. The learned semantic indexer can facilitate various downstream tasks, such as recommendation and retrieval. We conduct experiments on three tasks including recommendation, product search, and document retrieval on five datasets from various domains, where LMINDEXER outperforms competitive baselines significantly and consistently. Code is available at `https://anonymous.4open.science/r/ICLR24-submit-B2E7/`.

## 1 INTRODUCTION

In the context of information retrieval (IR), unique IDs are usually assigned to the documents, as doing so facilitates various downstream tasks including indexing and retrieval. For example, in the realm of e-commerce platforms, products are often tagged with distinctive product IDs (He & McAuley, 2016), and web passages are linked to specific URLs (Kousha & Thelwall, 2007). However, these document or item IDs are often randomly assigned, lacking the assurance of accurately encapsulating the underlying characteristics or content information of items and documents. This issue hinders the effective understanding, indexing, searching, and analysis of these items or documents based solely on their IDs. Thus, **semantic ID**, which is a sequence of discrete ID numbers that captures the semantic meaning of a document, has been proposed as an advanced unique ID to address this issue. The objective is to ensure that the initial set of semantic IDs captures the coarse-grained document semantics while the subsequent IDs delve into the details of its content in a hierarchical structure.

Recent research efforts (Tay et al., 2022; Wang et al., 2022; Rajput et al., 2023) have focused on acquiring semantic IDs through a self-supervised approach employing a two-step methodology. Generally, they first procure embeddings for documents using off-the-shelf text encoders, such as BERT (Devlin et al., 2019), under the assumption that these embeddings possess the capacity to encapsulate the semantic essence of documents for indexing purposes. They then employ specific techniques such as rq-VAE (Lee et al., 2022) or hierarchical clustering (Murtagh & Contreras, 2012) to derive semantic IDs for the documents, using the embeddings obtained as input. However, a notable issue arises due to the inherent mismatch between the distribution of embeddings in the latent space generated by text encoders and the expected distribution necessary for effective semantic indexing. Typically, the former exhibits a uniform distribution (Wang & Isola, 2020), while the latter requires a hierarchical structure to capture the coarse-grained to fine-grained semantics. Furthermore, each step of this process introduces potential information loss (Beaudry & Renner, 2012), as the embeddings may not faithfully preserve the entirety of the original document's semantics, and the second-stage methods may not produce flawless IDs.

To this end, we formulate the semantic ID learning task into a sequence-to-sequence fashion and propose to learn semantic IDs by *capturing the document's semantic representations and its hierar-*

*chical structure simultaneously*, with a generative language model, following (Raffel et al., 2020; Radford et al., 2019). However, developing such a generative language model-based method poses a formidable challenge, primarily rooted in two key aspects: 1) **Sequential discrete ID**: Semantic IDs, designed to capture the hierarchical semantics of documents, are sequentially structured. Initial IDs tend to encapsulate broad, coarse-grained semantics, while subsequent IDs delve into more refined, granular details. The inherent discreteness of these IDs adds complexity to end-to-end learning processes. 2) **Semantic supervision deficiency**: There's a conspicuous absence of supervisory signals to guide the specific allocation of semantic IDs to documents. It remains non-trivial to discern how semantically similar documents should be mapped to analogous semantic IDs. Addressing the two nuanced challenges requires the development of an advanced framework that is adept at unraveling and incorporating intricate semantic structures within documents. This ensures accurate and uniform allocation of semantic IDs, all while navigating the inherent limitations brought about by their discrete and sequential characteristics.

In pursuit of this goal, we introduce LMINDEXER, an innovative self-supervised approach designed to acquire semantic IDs directly from the input document with a generative language model, mastering the concurrent learning of the document's semantic representations and hierarchical structures. We tackle the challenge of *sequential discrete ID* by developing a semantic indexer capable of producing neural sequential discrete representations and designing a progressive training and contrastive learning paradigm. These designs adeptly encapsulate the hierarchical semantic intricacies of the input text within these IDs. In response to the *semantic supervision deficiency*, we employ a specialized reconstructor to rebuild the original text from the sequential discrete semantic ID representations acquired from the indexer via self-supervised learning. Our approach is grounded in the assumption that an effective semantic indexer should condense document-level semantics into these IDs, enabling a reconstructor to learn to accurately rebuild the original document from the obtained IDs. Following the self-supervised learning phase, the semantic indexer excels in producing semantic IDs for documents and can also undergo fine-tuning for various downstream tasks, including recommendation and retrieval.

To summarize, our main contributions are as follows:

- Conceptually, we formulate the problem of learning semantic IDs by capturing the document's semantic representations and its hierarchical structure simultaneously.

- Methodologically, we propose LMINDEXER, a self-supervised framework that contains a semantic indexer to generate semantic IDs and a reconstructor to reconstruct the original text from the IDs. The learned semantic indexer can be further fine-tuned on different downstream tasks.

- Empirically, we conduct experiments on three downstream tasks on five datasets from different domains, where LMINDEXER outperforms competitive baselines significantly and consistently.

## 2 RELATED WORK

**Self-supervised Learning with Language Models.** Training language models through self-supervision involves tuning the model without any labeled data, relying solely on the input text corpus itself. This concept has been extensively explored in existing literature (Devlin et al., 2019; Liu et al., 2019; Clark et al., 2020). BERT (Devlin et al., 2019) introduces two self-supervised training objectives, masked language modeling, and next sentence prediction. By training on vast text corpora, BERT demonstrates that the learned model could significantly enhance performance in downstream tasks. Liu et al. (2019) expands on this notion with RoBERTa, emphasizing the critical role of masked language modeling. In the information retrieval domain, SEED (Lu et al., 2021) proposes to pretrain the dense retriever with an attention-restricted decoder. CPDAE (Ma et al., 2022) introduces a contrastive pretraining approach to learn a discriminative autoencoder with a lightweight multilayer perception decoder. RetroMAE (Xiao et al., 2022) proposes a new retrieval-oriented pretraining paradigm based on Masked Auto-Encoder (MAE). However, most prior research has primarily focused on employing self-supervised learning to train language models for natural language understanding and dense retrieval. In contrast, this work explores the potential of self-supervised learning in utilizing language models as semantic indexers.

**Semantic indexer.** Semantic indexers (Van Den Oord et al., 2017; Lee et al., 2022; Esser et al., 2021) are initially introduced in computer vision, where they convert input images into a set of IDs capturing the essence of the original image. In Van Den Oord et al. (2017), an auto-encoder framework is proposed. The encoder learns discrete latent variables for input images, while the decoder reconstructs input from these discrete variables. Lee et al. (2022) enhances this with a residual quantizer for higher-quality semantic IDs. More recently, semantic IDs have been applied

Figure 1: The LMINDEXER self-supervised ID learning framework overview. The proposed semantic indexer includes a semantic ID encoder and several codebooks. During self-supervised learning, there is a reconstructor to reconstruct the input document from semantic ID representations.

to information retrieval tasks, such as document retrieval (Tay et al., 2022) and recommendations (Rajput et al., 2023). These IDs represent documents and are adopted in generative recommendation (Hua et al., 2023) and retrieval (Sun et al., 2023). Nevertheless, the development of these IDs highly relies on prior knowledge or supervision from the downstream tasks. Current self-supervised semantic indexing methods generally follow a two-step process. In the first step, an off-the-shelf text encoder (Devlin et al., 2019) encodes input documents and generates embedding representations for them. In the second step, either rq-VAE (Rajput et al., 2023) or hierarchical clustering (Tay et al., 2022; Wang et al., 2022) is employed to create IDs for documents based on the embeddings from the first step. Typically, there's a disparity between the distribution of embeddings in the latent space produced by text encoders and the expected distribution for semantic indexing. Furthermore, each stage incurs information loss (Beaudry & Renner, 2012). In this work, we introduce an innovative self-supervised approach designed to acquire semantic IDs directly from the input document with a generative language model, learning the document's semantic embeddings and its hierarchical structure simultaneously.

## 3 THE LMINDEXER FRAMEWORK

In this section, we present our LMINDEXER framework, which learns the document's semantic representations and its hierarchical structure simultaneously. In Section 3.1, we first introduce in detail how to design and train a generative language model-based semantic indexer (including a semantic encoder and codebooks) and tackle the sequential discrete ID and semantic supervision deficiency challenges. In Section 3.2, we discuss how to effectively optimize the LMINDEXER framework. In Section 3.3, we illustrate how to apply the learned document semantic IDs and the semantic indexer on downstream tasks. The overview of our proposed model is shown in Figure 1.

### 3.1 LEARNING SEMANTIC IDS WITH SEQUENTIAL DISCRETE AUTO-RECONSTRUCTION

Learning semantic IDs is challenging, given that semantic IDs are discrete and structured sequentially to represent the document's semantics hierarchically, and there is no semantic supervision to guide the training, *i.e.*, ground truth (document, semantic ID) pairs. To this end, we propose to learn semantic IDs as sequential discrete representations to capture the text semantics and train the semantic indexer with the self-supervised text reconstruction objective to tackle the semantic supervision deficiency. In the forthcoming sections, we employ bold notation to signify vectors, while non-bold notation is used to denote single values or units.

**Learning semantic IDs as neural sequential discrete representations.** The semantic indexer takes a document as the input and outputs its semantic ID that captures its semantic meaning. Therefore, learning a semantic indexer naturally formulates a text-to-text language model training problem. Following (Vaswani et al., 2017), we adopt an encoder-decoder Transformer architecture as the base model. Let $c_d^i$ denote the semantic ID of a document $d$ at position $i$. Given a document $d$ and its learned prefix ID $c_d^{<t} = c_d^1...c_d^{t-1}$ before position $t$, the semantic encoder will encode them and produce the latent vector representation $\boldsymbol{h}_d^t \in \mathcal{R}^D$ of $d$ at position $t$ as:

$$\boldsymbol{h}_d^t = \text{SemEnc}_\theta(d, c_d^{<t}) = \text{TransDecoder}(\text{TransEncoder}(d), c_d^{<t}). \quad (1)$$

Here TransEncoder is the Transformer encoder to capture the semantics of the input document and TransDecoder is the Transformer decoder designed to generate continuous sequential semantic ID hidden representations based on TransEncoder$(d)$ and $c_d^{<t}$. $D$ is the dimension of the hidden state.

The semantic indexer then maps the continuous hidden state $\boldsymbol{h}_d^t$ to a discrete ID $c_d^t$. At each ID position, we will maintain a codebook embedding matrix $\boldsymbol{E}^t \in \mathcal{R}^{K \times D}$, where $K$ is the codebook size. We have different codebook embedding matrices at each position to capture semantics of different

granularity. Each code embedding $e_j^t \in \mathcal{R}^D$ in $E^t$ corresponds to a semantic ID $j$ at position $t$. Based on $E^t$, the discrete semantic ID for document $d$ at $t$ is calculated by the dot-product look up:

$$P_s(c_d^t = j | c_d^{<t}, d) = \text{Softmax}_{e_j^t \in E^t}(h_d^t \cdot e_j^t), \qquad (2)$$

$$c_d^t = \text{argmax}_j P_s(c_d^t = j | c_d^{<t}, d). \qquad (3)$$

After this, document $d$ is represented by a sequence of semantic IDs $c_d = c_d^1 c_d^2 ... c_d^T$, corresponding to sequential discrete representations $\boldsymbol{c}_d = \boldsymbol{c}_d^1 \boldsymbol{c}_d^2 ... \boldsymbol{c}_d^T$, where $\boldsymbol{c}_d^t = E^t[c_d^t] \in \mathcal{R}^D$ and $T$ is ID length. The preliminary set of $c_d$ should predominantly encapsulate coarse-grained semantics, with successive IDs delving deeper into nuanced specifics of $d$. We will discuss how to capture the hierarchical structure by progressive training and contrastive learning in Section 3.2.

**Reconstructing document with sequential discrete semantic ID embeddings.** The document IDs are expected to capture the document-level semantics. As a result, high-quality document IDs should be able to be utilized to reconstruct the original document. To implement this intuition, we propose a Transformer reconstructor to perform document reconstruction.

The input of the reconstructor is the sequential discrete semantic ID representations $\boldsymbol{c}_d$ and the expected output is the document content $d$. Following Xiao et al. (2022), we consider providing some context hints $d_{\text{h}}$ and transfer the reconstruction into a masked token prediction style (Devlin et al., 2019). We randomly mask some tokens in $d$ and use $\boldsymbol{c}_d$ to decode those masked tokens together with $d_{\text{h}}$. To be specific, the reconstruction objective is calculated by

$$\mathcal{L}_{\text{recon}} = -\sum_d \sum_{w \in d \backslash d_{\text{h}}} \log P_{\text{recon}}(w | \boldsymbol{c}_d, d_{\text{h}}). \qquad (4)$$

Here $P_{\text{recon}}(w | \boldsymbol{c}_d, d_{\text{h}})$ is calculated by a shallow bidirectional Transformer (Trans) layer, where $\boldsymbol{c}_d$ is fed as the query channel input embeddings, and $\boldsymbol{d}_{\text{h}}$ (token embeddings correspond to $d_{\text{h}}$) are fed as key and value channel input embeddings in the multi-head self-attention. We adopt a shallow reconstructor which has limited reconstruction capability based only on the hints in order to force the semantic indexer to provide high-quality representations. The reconstruction is conducted as follows:

$$\boldsymbol{z}_w = \text{Recon}_\phi(\boldsymbol{c}_d, \boldsymbol{d}_{\text{h}}) = \sum_t \text{Trans}(q = \boldsymbol{c}_d^t, k = \boldsymbol{d}_{\text{h}}, v = \boldsymbol{d}_{\text{h}})$$
$$P_{\text{recon}}(w | \boldsymbol{c}_d, d_{\text{h}}) = \text{softmax}(\boldsymbol{W} \boldsymbol{z}_w) \qquad (5)$$

where $\boldsymbol{W}$ is the token embedding matrix. However, directly adopting the reconstruction objective with $\boldsymbol{c}_d$ as input to the reconstructor will not optimize the semantic encoder. Since the codebook look-up in Eq.(2) is a hard/discrete operation, the reconstruction objective backpropagation gradients will flow to the embeddings in the codebook rather than to the parameters in the semantic encoder. To this end, we propose to approximate the argmax operation similar to (Jang et al., 2016) as follows:

$$\hat{\boldsymbol{c}}_d^t = \begin{cases} \arg\max_{e_j^t \in E^t} h_d^t \cdot e_j^t & \text{forward pass.} \\ \sum_{e_j^t \in E^t} \frac{\exp(h_d^t \cdot e_j^t)}{\sum_{e_j^t \in E^t} \exp(h_d^t \cdot e_j^t)} e_j^t & \text{backward pass.} \end{cases} \qquad (6)$$

In the forward pass, we still adopt the argmax($\cdot$) hard operation; while in the backward pass, the selected semantic embedding becomes a weighted average of the codebook embeddings, to enable gradients to flow to $h_d^t$ and finally to the parameters in the semantic encoder. In our implementation, we achieve this by adopting the *stop gradient* operator (Van Den Oord et al., 2017). The reconstruction is then conducted by

$$\boldsymbol{z}_w = \text{Recon}_\phi(\hat{\boldsymbol{c}}_d^t, \boldsymbol{d}_{\text{h}}) = \sum_t \text{Trans}(q = \hat{\boldsymbol{c}}_d^t, k = \boldsymbol{d}_{\text{h}}, v = \boldsymbol{d}_{\text{h}}) \qquad (7)$$

## 3.2 TRAINING SELF-SUPERVISED SEMANTIC INDEXER

**Progressive training.** To optimize the semantic indexer and obtain semantic IDs in an auto-regressive way, we adopt the progressive training scheme similar to Sun et al. (2023). The entire learning process consists of $T$ learning steps, each corresponding to a specific semantic ID $c_d^t$ being learned and optimized at position $t$ within the range of $[T]$. Additionally, at each step $t$, both the ID $c_d^t$ and the model parameters associated with generating $c_d^t$ are updated, while previously generated IDs $c_d^{<t}$

remain unchanged. The reconstruction objective in $t$-step is shown as:

$$\mathcal{L}_{\text{recon}}^t = -\sum_d \sum_{w \in d \setminus d_{\text{h}}^t} \log P_{\text{recon}}(w | \boldsymbol{c}_d^{\leq t}, d_{\text{h}}^t). \tag{8}$$

Here $d_{\text{h}}^t$ is the hints provided for learning ID on position $t$. We will gradually reduce the amounts of hints $d_{\text{h}}^t$ as $t$ increases to inject new knowledge into the new IDs, and finally contribute to a hierarchical, coarse-to-fine-grained semantic ID learning.

**Contrastive loss.** The reconstruction objective in Eq.(8) can force the semantic IDs to capture document-level semantics. However, only optimizing the objective can lead to the case where similar documents sharing $c_d^{\leq t}$ also have the same $c_d^t$. To alleviate this issue, we propose a contrastive objective to promote distinction between documents that previously shared the same prefix, enabling the model to discern finer-grained hierarchical relationships between documents:

$$\mathcal{L}_{\text{contrastive}}^t = -\sum_d \log \frac{\exp(\boldsymbol{h}_d^t \cdot \boldsymbol{h}_d^t)}{\exp(\boldsymbol{h}_d^t \cdot \boldsymbol{h}_d^t) + \sum_{c_{d'}^{\leq t} = c_d^{\leq t}} \exp(\boldsymbol{h}_d^t \cdot \boldsymbol{h}_{d'}^t)}. \tag{9}$$

The contrastive objective can help push $\boldsymbol{h}_d^t$ of documents sharing the same $c_d^{\leq t}$ away in the $t$-th latent space and force them to obtain diverse $c_d^t$, finally contributing to higher codebook utilization.

**Commitment loss.** In addition, when learning the document semantic IDs for position $t$, it is important that the semantic indexer should remember the IDs that are already learned before position $t$. To this end, we add a commitment loss as:

$$\mathcal{L}_{\text{commitment}}^t = -\sum_d \sum_{j < t} \log P_s(c_d^j | d, c_d^{<j}). \tag{10}$$

We optimize our model at step $t$ based on a combination of the three losses proposed above:

$$\min_{\theta, \phi, \boldsymbol{E}^t} \mathcal{L}^t = \mathcal{L}_{\text{recon}}^t + \mathcal{L}_{\text{contrastive}}^t + \mathcal{L}_{\text{commitment}}^t. \tag{11}$$

However, we empirically find that directly pursuing optimization of the above objective is suboptimal as the model would encounter two forms of collapse: reconstructive collapse and posterior collapse.

**Reconstructor collapse.** It refers to the case when the reconstructor is performing badly and misguides the semantic indexer. It could happen when the reconstructor is under-trained and back-propagates noisy gradients to the semantic indexer (Xiao et al., 2022). This problem appears in our framework since the reconstructor is randomly initialized. We solve this problem by first fixing the semantic encoder component and warming up the parameters in the reconstructor:

$$\min_\phi \mathcal{L}_{\text{recon}}^0 = -\sum_d \sum_{w \in d \setminus d_{\text{h}}^0} \log P_{\text{recon}}(w | d_{\text{h}}^0). \tag{12}$$

**Posterior collapse.** It refers to the case when the information provided by the semantic indexer is weak and noisy for the reconstructor, thus is not utilized in the reconstruction (He et al., 2018). This problem appears in our framework since the representations the reconstructor receives from the semantic indexer are approximated by codebook embeddings (find in Eq.(2)) which are randomly initialized. We solve this problem by first training the auto-reconstruction framework without the $t$-th codebook at each step $t$:

$$\min_{\theta, \phi} \mathcal{L}^t, \quad \boldsymbol{z}_w = \text{Recon}_\phi(\boldsymbol{c}_d^{<t}, \boldsymbol{h}_d^t, d_{\text{h}}^t) \tag{13}$$

and initialize the codebook embeddings with a good initialization (*e.g.,* Kmeans of $\{\boldsymbol{h}_d^t\}_d$) from the trained semantic encoder before optimizing Eq.(11). More detailed studies on the two collapses can be found in Section 4.3. A detailed training procedure can be found in Appendix A.2.

### 3.3 FINETUNING SEMANTIC INDEXER ON DOWNSTREAM TASKS

After we obtain a self-supervised learned semantic ID indexer, it can then be directly utilized to generate semantic IDs for documents both seen and unseen in the training corpus. Meanwhile, the semantic indexer can also be finetuned on downstream tasks which take text as input and expect document IDs as output, *e.g.*, recommendation (user history interaction text as input and next item ID as output) and retrieval (query as input and document ID as output) as shown in Figure 2. To be specific, given a set of downstream task samples $\mathcal{D} = \{(q, d)\}$ where $q$ is the input text and $d$ is the

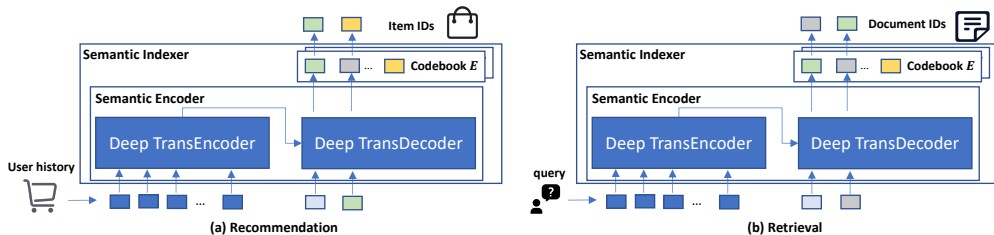

Figure 2: LMINDEXER can be fine-tuned on downstream tasks including recommendation (user history as input and item ID as output) and retrieval (query as input and document ID as output).

expected output documents, we first obtain the semantic IDs $c_d$ corresponding to $d$ with the learned semantic indexer. We then finetuned the semantic index on this task with $\mathcal{D} = \{(q, c_d)\}$ as follows:

$$\mathcal{L}_{\text{downstream}} = - \sum_{(q, c_d) \in \mathcal{D}} \sum_{j \leqslant T} \log P_s(c_d^j | q, c_d^{<j}). \tag{14}$$

In the inference stage, we conduct constrained beam search decoding with a prefix tree (Wang et al., 2022), which in turn only generates valid document IDs.

# 4 EXPERIMENTS: LEARNING SELF-SUPERVISED SEMANTIC ID

## 4.1 EXPERIMENTAL SETUP

**Datasets.** We conduct semantic ID learning experiments on product corpus from three domains in Amazon review dataset (He & McAuley, 2016): Amazon-Beauty, Amazon-Sports, and Amazon-Toys, as well as the document corpus from Natural Question (Kwiatkowski et al., 2019) and MS MACRO (Nguyen et al., 2016). For items in Amazon, their title, description, and features are concatenated as textual information; while for documents in NQ and MACRO, their content is treated as textual features. The statistics of the datasets can be found in Table 6 and Table 7 in the Appendix.

**Implementation details.** In our experiments, we use T5-base (Raffel et al., 2020) as the base model for our semantic indexer. The reconstructor is a 1-layer Transformer. The length of the semantic IDs is set as $T = 3$. We have different codebook embeddings initialized for different positions $t$ and the size of the codebook is set to be in $\{512, 5120, 51200\}$ depending on the size of the document corpus. We optimize the model with AdamW and search the learning rate in $\{1e\text{-}3, 2e\text{-}3, 5e\text{-}3\}$. The training epochs are set to be 30, 10, and 5 for Amazon datasets, NQ, and MS MACRO respectively. More information on implementation details can be found in Appendix A.3.

## 4.2 SEMANTIC ID QUALITY ANALYSIS

**Baselines.** We compare with two self-supervised semantic indexer methods mentioned in previous works: rq-VAE indexer (Rajput et al., 2023) and hierarchical clustering (HC) indexer (Tay et al., 2022). Both methods adopt the two-step paradigm: 1) derive embeddings with the off-the-shelf text encoder (Devlin et al., 2019); 2) obtain IDs based on the embeddings with rq-VAE (Lee et al., 2022) or hierarchical clustering (Murtagh & Contreras, 2012).

**Quantitative results.** We conduct a quantitative evaluation to measure the quality of self-supervised learned semantic IDs. Semantic IDs of high quality should capture the semantic similarity between documents. In other words, documents of similar IDs should be of similar semantics. In this section, we calculate the AMI (Vinh et al., 2009)

Table 1: ID quantitative study (AMI).

| Model | Beauty | Sports | Toys |
|---|---|---|---|
| rq-VAE indexer | 0.2654 | 0.2774 | 0.3154 |
| HC indexer | 0.2428 | 0.2387 | 0.2729 |
| LMINDEXER | **0.3563** | **0.4163** | **0.3536** |

score between item semantics IDs and ground truth item category (which can serve as ground truth semantics) in Amazon datasets. The results are shown in Table 1. From the result, LMINDEXER outperforms baseline methods consistently, which demonstrates that the IDs learned by LMINDEXER are more semantic-indicative.

**Qualitative results.** We conduct case studies on the learned semantic IDs from LMINDEXER on Amazon-Toys. We randomly select two semantic ID prefixes and print out the shared keywords of the items corresponding to the two IDs. The results are shown in Figure 3, where we can find that items in

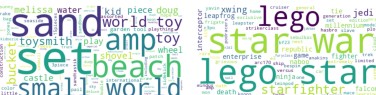

(a) Semantic ID 1   (b) Semantic ID 2
Figure 3: Qualitative study.

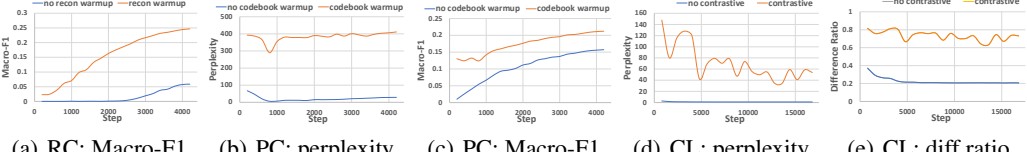

(a) RC: Macro-F1     (b) PC: perplexity     (c) PC: Macro-F1     (d) CL: perplexity     (e) CL: diff ratio

Figure 4: Semantic indexer training analysis on Amazon-sports. x-axis denotes the training step and y-axis denotes the evaluation metrics. Reconstructor collapse analysis (a): The reconstructor suffers from low reconstruction Macro-F1 without reconstructor warm-up (blue). Posterior collapse analysis (b,c): The semantic indexer suffers from generating homogeneous and meaningless IDs (low perplexity), and results in low reconstruction Macro-F1, without encoder and codebook warm-up (blue). Contrastive learning analysis (d,e): Documents sharing prefix ID tend to have similar next position ID (low diff ratio) and low diversity (low perplexity) without contrastive objective (blue).

the two groups are related to beach/sand set toys, and star war toys respectively. This demonstrates that the IDs acquired through LMIndexer convey a higher degree of semantic relevance.

### 4.3 TRAINING STUDY

In this section, we study the optimization process (reconstructor collapse, posterior collapse, and contrastive loss discussed in Sec 3.2) of our semantic indexer from two perspectives: reconstruction quality and semantic ID diversity. We serve token reconstruction Macro-F1 (Opitz & Burst, 2019) and semantic ID perplexity of all the documents (Horgan, 1995) as the main evaluation metrics. A high-quality semantic indexer should contribute to a high reconstruction quality (high Macro-F1) and a high semantic ID diversity (high perplexity). We conduct model studies on Amazon-sports shown in Figure 4 and have the following findings: 1) **Reconstructor collapse**: The reconstruction Macro-F1 is low without reconstructor warm-up, shown in Figure 4(a). In this case, the reconstructor suffers from low reconstruction capability and cannot provide meaningful signals to train the semantic indexer. 2) **Posterior collapse**: The semantic ID perplexity is low without semantic encoder and codebook warm-up, shown in Figure 4(b). This indicates that the semantic indexer fails to provide diverse and meaningful signals to the reconstructor and thus results in low reconstruction macro-F1 in Figure 4(c). 3) **Contrastive loss**: We propose the contrastive loss in Section 3.2 to push documents sharing the same semantic ID prefix to obtain different IDs at the current step. We show the effectiveness of this design in Figure 4(d)(e). Difference ratio (diff ratio) refers to the ratio of # (document pairs sharing the same ID prefix but having different current step IDs) / # (document pairs sharing the same ID prefix). From the result, we can find that the difference ratio is high with contrastive loss, which makes the semantic IDs more distinguishable for different documents.

## 5 EXPERIMENTS: DOWNSTREAM TASKS

### 5.1 SEQUENTIAL RECOMMENDATION

**Task definitions.** Given the historical data of user $u$'s interacted items $I_u$, the task is to predict which next item $v$ the user will interact with in the future.

**Datasets.** We conduct experiments on three domains from Amazon review dataset (He & McAuley, 2016): Amazon-Beauty, Amazon-Sports, and Amazon-Toys. We keep the users and items that have at least 5 interactions in their history in the Amazon review dataset. We treat the last interacted item by each user as the testing sample, the last second interacted item as the validation sample, and the previous items as training samples. The statistics of the datasets can be found in Appendix Table 6.

**Baselines.** We compare our method with both popular sequential recommendation models including HGN (Ma et al., 2019), GRU4Rec (Hidasi et al., 2016), BERT4Rec (Sun et al., 2019) and FDSA (Zhang et al., 2019), as well as generative recommendation methods with semantic IDs (Rajput et al., 2023; Tay et al., 2022): rq-VAE indexer and hierarchical clustering (HC) indexer.

**Implementation details.** For generative recommendation methods (rq-VAE indexer, hierarchical clustering indexer, and LMINDEXER), we concatenate the textual information (title & description) of the user's previously interacted items, serve it as the input text into the generative language model and ask the model to generate the ID for next item. The baselines are using the same T5-base checkpoint. We train all the compared generative recommendation methods for 10,000 steps with the learning rate searched in {1e-2, 1e-3, 1e-4}. The batch size is set to 32, the maximum input text length is set to 1024 and all experiments are run on an 8 A100 40G machine. The number of beams for beam search is set to 20.

Table 2: Next item recommendation.

| | Amazon-Beauty | | Amazon-Sports | | Amazon-Toys | |
| Model | Recall@5 | NDCG@5 | Recall@5 | NDCG@5 | Recall@5 | NDCG@5 |
| --- | --- | --- | --- | --- | --- | --- |
| HGN | 0.0325 | 0.0206 | 0.0189 | 0.0120 | 0.0321 | 0.0221 |
| GRU4Rec | 0.0164 | 0.0099 | 0.0129 | 0.0086 | 0.0097 | 0.0059 |
| BERT4Rec | 0.0203 | 0.0124 | 0.0115 | 0.0075 | 0.0116 | 0.0071 |
| FDSA | 0.0267 | 0.0163 | 0.0182 | 0.0122 | 0.0228 | 0.0140 |
| rq-VAE indexer | 0.0136 | 0.0086 | 0.0067 | 0.0040 | 0.0084 | 0.0055 |
| HC indexer | 0.0129 | 0.0078 | 0.0076 | 0.0050 | 0.0082 | 0.0054 |
| LMINDEXER | **0.0415** | **0.0262** | **0.0222** | **0.0142** | **0.0404** | **0.0268** |

Table 3: Product search.

| | Amazon-Beauty | | Amazon-Sports | | Amazon-Toys | |
| Model | NDCG@5 | MAP@5 | NDCG@5 | MAP@5 | NDCG@5 | MAP@5 |
| --- | --- | --- | --- | --- | --- | --- |
| bm25 | 0.2490 | 0.2152 | 0.1898 | 0.1581 | 0.2085 | 0.1760 |
| Dual Encoder | 0.2565 | 0.2096 | 0.2556 | 0.2223 | 0.2805 | 0.2420 |
| rq-VAE indexer | 0.2710 | 0.2469 | 0.2606 | 0.2354 | 0.2511 | 0.2287 |
| HC indexer | 0.2172 | 0.1959 | 0.1979 | 0.1812 | 0.2379 | 0.2156 |
| LMINDEXER | **0.3187** | **0.2888** | **0.2870** | **0.2607** | **0.2865** | **0.2592** |

**Main result.** The performance comparisons of different methods are shown in Table 2. From the results, we can find that: 1) LMINDEXER performs consistently better than all the baseline methods on all datasets. 2) Although other generative recommendation methods employing semantic IDs share a similar encoding approach with LMINDEXER, their performance is hampered by limitations in the quality of their semantic indexers and item IDs.

**Semantic ID length study.** In this section, we analyze how the length of the semantic IDs affects the downstream recommendation performance. We conduct experiments with the length of item semantic IDs to be 1, 2, and 3. The results on the Amazon-Beauty dataset are shown in Figure 5. From the result, we can find that the model performance increases as the semantic ID length increases. The result is intuitive, since the longer the semantic ID is, the more semantic information it can contain.

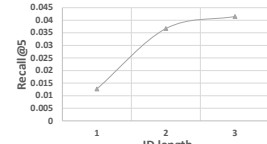

Figure 5: Semantic ID length study on Amazon-beauty.

## 5.2 PRODUCT SEARCH

**Task definitions.** Given a query $q$ provided by a user, retrieve relevant item $v$ he/she will be interested in from the product collection.

**Datasets.** We conduct experiments on three domains from the Amazon product search dataset (Reddy et al., 2022): Amazon-Beauty, Amazon-Sports, and Amazon-Toys. To verify if the learned semantic IDs can generalize to different downstream tasks, we keep the product corpus in the three domains the same as those in Section 5.1. We select the queries in the original product search dataset (Reddy et al., 2022) which correspond to ground truth products in the product corpus and use the original train/test split. The statistics of the datasets can be found in Appendix Table 6.

**Baselines.** We compare our method with traditional retrieval method bm25 (Robertson et al., 2009), dual encoder DPR (Karpukhin et al., 2020), as well as generative retrieval methods with semantic IDs (Rajput et al., 2023; Tay et al., 2022): rq-VAE indexer, and hierarchical clustering (HC) indexer.

**Implementation Details.** For generative retrieval methods (rq-VAE indexer, hierarchical clustering indexer, and LMINDEXER), we serve the query as the input text into the generative language model and ask the model to generate the ID for the relevant items. All baselines initially load the same T5-base checkpoint. We train all the compared generative retrieval methods for 10,000 steps with the learning rate searched in {1e-2, 1e-3, 1e-4}. The batch size is set to 32, the maximum input text length is set to 1024 and all experiments are run on an 8 A100 40G machine. The number of beams for beam search is set to 20.

**Main result.** The performance comparisons of different methods are shown in Table 3. From the results, we can find that: 1) LMINDEXER performs consistently better than all the baseline methods

Table 5: Document retrieval.

| Model | NQ320k | | TREC-DL 1M | | MACRO 1M |
| | Recall@1 | Recall@10 | Recall@10 | NDCG@10 | MRR@10 |
|---|---|---|---|---|---|
| bm25 | 0.2970 | 0.6030 | 0.2756 | 0.2995 | 0.3144 |
| Dual Encoder | 0.5360 | 0.8300 | 0.3612 | 0.3941 | **0.5561** |
| rq-VAE indexer | *0.6480* | *0.8322* | 0.4199 | *0.4579* | 0.5159 |
| HC indexer | 0.6439 | 0.8213 | *0.4265* | 0.4571 | 0.5126 |
| LMINDEXER | **0.6631** | **0.8589** | **0.4519** | **0.4695** | *0.5485* |

on all datasets. 2) The dual encoder model DPR is a strong approach, outperforming semantic indexer baselines (rq-VAE indexer and hierarchical clustering indexer) in many cases.

**Zero-shot study.** We conduct zero-shot product search experiments on the Amazon beauty domain to test if the semantic indexer finetuned on downstream tasks can generalize to items that are not seen during semantic index self-supervised training and downstream finetuning. The results are shown in Table 4. From the results, we can find

Table 4: Zero-shot study.

| Model | Recall@50 | Recall@100 |
|---|---|---|
| rq-VAE indexer | 0.0000 | 0.0105 |
| HC indexer | 0.0000 | 0.0070 |
| LMINDEXER | **0.0455** | **0.0524** |

that compared with other semantic indexer methods, LMINDEXER can generalize better to unseen documents, demonstrating its strong semantic capturing capability.

## 5.3 DOCUMENT RETRIEVAL

**Task definitions.** Given a query $q$, retrieve relevant documents $v$ from a document corpus.

**Datasets.** We conduct experiments on Natural Question (Kwiatkowski et al., 2019) and MS MACRO (Nguyen et al., 2016). MS MACRO dev and TREC-DL (Craswell et al., 2020) are used as the evaluation set for MS MACRO. Following Pradeep et al. (2023), we construct an MS MACRO-1M by extracting a 1 million document subset from the original collection and keeping the original training and validation labels. We merge the TREC-DL 2019 and TREC-DL 2020 datasets, keep the documents appearing in MACRO 1M, and develop a larger TREC-DL dataset. The detailed statistics of all the datasets can be found in Appendix Table 7.

**Baselines.** We compare our method with traditional retrieval method bm25 (Robertson et al., 2009), dual encoder DPR (Karpukhin et al., 2020), as well as generative retrieval methods with semantic IDs (Rajput et al., 2023; Tay et al., 2022): rq-VAE indexer and hierarchical clustering indexer.

**Implementation Details.** For generative retrieval with semantic ID methods (rq-VAE indexer, hierarchical clustering indexer, and LMINDEXER), we serve the query as the input text into the semantic indexer and ask the model to generate the ID for the relevant documents. Following (Wang et al., 2022), we use docT5query (Nogueira et al., 2019) to generate pseudo queries for each document in NQ and MS MACRO for training augmentation. The number of pseudo queries for each document is set to 15 and 20 respectively. We train all the compared generative retrieval methods for 250,000 and 500,000 steps in NQ and MS MACRO respectively, with the learning rate searched in {5e-4, 1e-3, 5e-3}. The batch size is set to 256, the maximum input text length is set to 32 and all experiments are run on an 8 A100 40G machine. The number of beams for beam search is set to 20. All baselines initially load the same T5-base checkpoint.

**Main result.** The performance comparisons of different methods are shown in Table 5. From the results, we can find that: 1) LMINDEXER performs consistently better than all the baseline methods except on MACRO 1M. 2) In the large corpus dataset MACRO 1M, the dual encoder method, *i.e.*, DPR, is still the best choice, leaving room for better semantic indexer methods to be developed.

## 6 CONCLUSIONS

In this paper, we introduce LMINDEXER, a self-supervised framework to learn semantic IDs with a generative language model, learning the document's discrete semantic embeddings and its hierarchical structure simultaneously. We address the challenge of sequential discrete ID by introducing a semantic indexer capable of generating neural discrete representations with progressive training and contrastive learning. In response to the semantic supervision deficiency, we propose to train the model using a self-supervised objective focused on reconstructing documents. The learned semantic indexer can be fine-tuned for various downstream tasks, such as recommendation and retrieval. We conduct experiments across three tasks including recommendation, product search, and document retrieval, using five datasets from diverse domains, where LMINDEXER outperforms competitive baselines significantly and consistently.

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

# A  APPENDIX

## A.1  DATASETS

For recommendation and product search, we conduct experiments on three domains from the Amazon review dataset (He & McAuley, 2016): Amazon-Beauty, Amazon-Sports, and Amazon-Toys. For recommendation, we keep the users and items with at least 5 interactions in their history in the Amazon review dataset. We treat the last interacted item by each user as the testing sample, the last second interacted item as the validation sample, and the previous items as training samples. For product search, to verify if the learned semantic IDs can be generalized to different downstream tasks, we keep the product corpus in the three domains the same as those in the recommendation experiments. We keep the queries in the original product search dataset (Reddy et al., 2022) which correspond to ground truth products in the product corpus. We use the original train/test split and randomly select 1/8 queries from the training set to be the validation set.

The statistics of the recommendation and product search datasets can be found in Table 6.

Table 6: Dataset Statistics

| Dataset | # Items | # Users | # Rec history (train/dev/test) | # Search query (train/dev/test) | # Search labels (train/dev/test) |
|---|---|---|---|---|---|
| Amazon-Beauty | 12,101 | 22,363 | 111,815 / 22,363 / 22,363 | 1,049 / 150 / 338 | 1,907 / 268 / 582 |
| Amazon-Sports | 18,357 | 35,598 | 177,990 / 35,598 / 35,598 | 1,299 / 186 / 443 | 2,209 / 311 / 764 |
| Amazon-Toys | 11,924 | 19,412 | 97,060 / 19,412 / 19,412 | 1,010 / 145 / 351 | 1,653 / 250 / 594 |

For document retrieval, we conduct experiments on Natural Question (NQ) (Kwiatkowski et al., 2019) and MS MACRO (Nguyen et al., 2016). For NQ, we keep the original training and testing labels and put all the documents together to form the text corpus. For MS MACRO, following Pradeep et al. (2023), we construct an MS MACRO-1M by extracting a 1 million document subset from the original collection and keeping the original training and validation labels. For TREC-DL, we merge the TREC-DL 2019 and TREC-DL 2020 datasets and keep the documents appearing in MACRO 1M. MS MACRO dev and TREC-DL Craswell et al. (2020) are used as the evaluation set for MS MACRO.

The statistics of the document retrieval datasets can be found in Table 7.

Table 7: Dataset Statistics

| Dataset | # Documents | # Query (train/test) | # Search labels (train/test) |
|---|---|---|---|
| NQ320k | 109,739 | 307,373 / 7,830 | 307,373 / 7,830 |
| MACRO 1M | 1,000,000 | 502,939 / 6,980 | 532,751 / 7437 |
| TREC-DL 1M | 1,000,000 | 502,939 / 93 | 532,751 / 1,069 |

## A.2  SUMMARY OF LMINDEXER'S SELF-SUPERVISED ID LEARNING PROCEDURE

## A.3  IMPLEMENTATION DETAILS

In self-supervised semantic indexer training, we use T5-base (Raffel et al., 2020) as our base model. The length of the semantic IDs is set as $T = 3$. The final position is added to distinguish documents sharing the first two position ID prefixes. For $t = 1$ and $t = 2$, we provide 50% hints and 30% hints for reconstruction respectively. We have different codebook embeddings initialized for different positions $t$ and the size of the codebook is set to be in {512, 5,120, 51,200} depending on the size of the document corpus. We optimize the model with AdamW and search the learning rate in {1e-3, 2e-3, 5e-3}. The training epochs are set to be 30, 10, and 5 for Amazon datasets, NQ, and MS MACRO respectively. The hyper-parameter configuration for self-supervised semantic indexer training can be found in Table 8.

In the downstream recommendation task, for generative recommendation methods with semantic IDs (rq-VAE indexer, hierarchical clustering indexer, and LMINDEXER), we concatenate the textual information (title & description) of the user's previously interacted items, serve it as the input text into the generative language model and ask the model to generate the ID for next item. The baselines are using the same T5-base checkpoint. We train all the compared generative recommendation methods

---

**Algorithm 1:** Self-supervised ID Learning Procedure of LMINDEXER

**Input** : The document corpus $\{d\}$.

**Output** : The semantic IDs $\{c_d\}$ of the documents $\{d\}$. A semantic indexer SemIndexer$(\cdot)$ which contains a semantic encoder SemEnc$_\theta(\cdot)$ and codebooks $\{\boldsymbol{E}^t\}_t$. A reconstruction model Recon$_\phi(\cdot)$.

**begin**

    // initialize semantic encoder

    SemEnc$_\theta(\cdot) \leftarrow$ T5-base ;

    // reconstructor warm up

    $\min_\phi \mathcal{L}_{\text{recon}}^0 = -\sum_d \sum_{w \in d \backslash d_h^0} \log P_{\text{recon}}(w|d_{\text{h}}^0)$ ;

    **for** $t = 1, ..., T$ **do**

        // semantic encoder & codebook warm up

        $\boldsymbol{h}_d^t \leftarrow$ SemEnc$_\theta(d, c_d^{<t})$ ;

        $\boldsymbol{z}_w \leftarrow$ Recon$_\phi(q = \{\boldsymbol{c}_d^{<t}, \boldsymbol{h}_d^t\}, k = \boldsymbol{d}_{\text{h}}^t, v = \boldsymbol{d}_{\text{h}}^t)$ ;

        $\min_{\theta,\phi} \mathcal{L}^t = \mathcal{L}_{\text{recon}}^t + \mathcal{L}_{\text{contrastive}}^t + \mathcal{L}_{\text{commitment}}^t$ ;

        $\boldsymbol{h}_d^t \leftarrow$ SemEnc$_\theta(d, c_d^{<t})$ ;

        $\boldsymbol{E}^t \leftarrow$ KMeans$(\{\boldsymbol{h}_d^t\})$ ;

        // whole framework training

        $\boldsymbol{z}_w \leftarrow$ Recon$_\phi(q = \{\boldsymbol{c}_d^{<t}, \hat{\boldsymbol{c}}_d^t\}, k = \boldsymbol{d}_{\text{h}}^t, v = \boldsymbol{d}_{\text{h}}^t)$ ;

        $\min_{\theta,\phi,\boldsymbol{E}^t} \mathcal{L}^t = \mathcal{L}_{\text{recon}}^t + \mathcal{L}_{\text{contrastive}}^t + \mathcal{L}_{\text{commitment}}^t$ ;

        $c_d^t \leftarrow \arg\max_j P_s(c_d^t = j|c_d^{<t}, d)$ ;

    **end**

    **return** $\{c_d\}$, *SemIndexer*$(\cdot)$ ;

**end**

Table 8: Hyper-parameter configuration for self-supervised semantic ID learning.

| Parameter | Amazon-Beauty | Amazon-Sports | Amazon-Toys | NQ | MACRO-1M |
|---|---|---|---|---|---|
| Optimizer | Adam | Adam | Adam | Adam | Adam |
| Adam $\epsilon$ | 1e-6 | 1e-6 | 1e-6 | 1e-6 | 1e-6 |
| Adam $(\beta_1, \beta_2)$ | (0.9, 0.999) | (0.9, 0.999) | (0.9, 0.999) | (0.9, 0.999) | (0.9, 0.999) |
| Batch size | 128 | 128 | 128 | 128 | 128 |
| Max epochs | 30 | 30 | 30 | 10 | 5 |
| Max sequence length | 512 | 512 | 512 | 512 | 128 |
| ID length | 3 | 3 | 3 | 3 | 3 |
| Codebook size | 512 | 512 | 512 | 5120 | 51200 |
| Hint ratio | 50%, 30% | 50%, 30% | 50%, 30% | 50%, 30% | 50%, 30% |
| Learning rate | | | searched in {1e-3, 2e-3, 5e-3} | | |
| Backbone LM | | | T5-base | | |

for 10,000 steps with the learning rate searched in {1e-2, 1e-3, 1e-4}. The batch size is set to be 32, the maximum input text length is set to be 1024 and all experiments are run on an 8 A100 40G machine. The number of beams for beam search is set to 20. The hyper-parameter configuration for generative recommendation training can be found in Table 9.

In the downstream product search task, for generative retrieval methods with semantic IDs (rq-VAE indexer, hierarchical clustering indexer, and LMINDEXER), we serve the query as the input text into the generative language model and ask the model to generate the ID for the relevant items. All baselines initially load the same T5-base checkpoint. We train all the compared generative retrieval methods for 10,000 steps with the learning rate searched in {1e-2, 1e-3, 1e-4}. The batch size is set to 32, the maximum input text length is set to be 1024 and all experiments are run on an 8 A100 40G machine. The number of beams for beam search is set to 20. The hyper-parameter configuration for generative product search training can be found in Table 10.

In the downstream document retrieval task, for generative retrieval methods with semantic IDs (rq-VAE indexer, hierarchical clustering indexer, and LMINDEXER), we serve the query as the input text into the semantic indexer and ask the model to generate the ID for the relevant documents. Following (Wang et al., 2022), we use docT5query (Nogueira et al., 2019) to generate pseudo queries for each document in NQ and MS MACRO for training augmentation. The number of pseudo queries for each document is set to be 15 and 20 respectively. We train all the compared generative retrieval methods for 250,000 and 500,000 steps in NQ and MS MACRO respectively, with the learning rate searched in {5e-4, 1e-3, 5e-3}. The batch size is set to 2048, the maximum input text length is set to 32 and all experiments are run on an 8 A100 40G machine. The number of beams for beam search is set to

Table 9: Hyper-parameter configuration for generative recommendation.

| Parameter | Amazon-Beauty | Amazon-Sports | Amazon-Toys |
|---|---|---|---|
| Optimizer | Adam | Adam | Adam |
| Adam $\epsilon$ | 1e-6 | 1e-6 | 1e-6 |
| Adam $(\beta_1, \beta_2)$ | (0.9, 0.999) | (0.9, 0.999) | (0.9, 0.999) |
| Batch size | 32 | 32 | 32 |
| Max steps | 10,000 | 10,000 | 10,000 |
| Max sequence length | 1024 | 1024 | 1024 |
| Bean size | 20 | 20 | 20 |
| Learning rate | searched in {1e-2, 1e-3, 1e-4} | | |
| Backbone LM | T5-base | | |

Table 10: Hyper-parameter configuration for generative product search.

| Parameter | Amazon-Beauty | Amazon-Sports | Amazon-Toys |
|---|---|---|---|
| Optimizer | Adam | Adam | Adam |
| Adam $\epsilon$ | 1e-6 | 1e-6 | 1e-6 |
| Adam $(\beta_1, \beta_2)$ | (0.9, 0.999) | (0.9, 0.999) | (0.9, 0.999) |
| Batch size | 32 | 32 | 32 |
| Max steps | 10,000 | 10,000 | 10,000 |
| Max sequence length | 1024 | 1024 | 1024 |
| Bean size | 20 | 20 | 20 |
| Learning rate | searched in {1e-2, 1e-3, 1e-4} | | |
| Backbone LM | T5-base | | |

20. All baselines initially load the same T5-base checkpoint. The hyper-parameter configuration for generative document retrieval training can be found in Table 11.

### A.4 SEMANTIC ID LENGTH STUDY

In this section, we analyze how the length of the semantic IDs affects the downstream recommendation performance. We conduct experiments with the length of item semantic IDs to be 1, 2, and 3. The results on the Amazon-Beauty, Amazon-Sports, and Amazon-Toys datasets are shown in Figure 6. From the result, we can find that the model performance increases as the semantic ID length increases. The result is intuitive, since the longer the semantic ID is, the more semantic information it can contain.

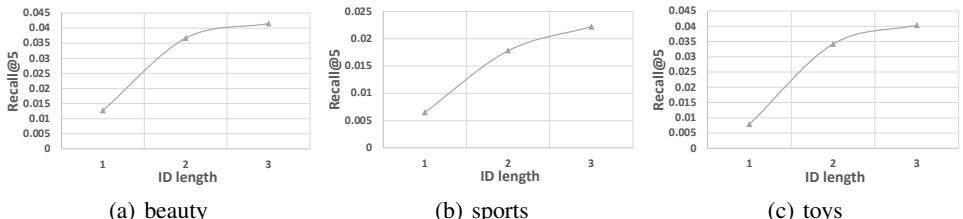

| (a) beauty | (b) sports | (c) toys |
|---|---|---|

Figure 6: Semantic ID length study on recommendation.

### A.5 COMPARISON WITH SEAL

We add experiments to compare our method with SEAL (Bevilacqua et al., 2022) on Amazon product search datasets and NQ320k dataset. SEAL is an autoregressive search engine that uses Ngrams as document identifiers. The results are shown in Table 12. From the result, we can find that our method outperforms SEAL significantly. The main reason is that after the self-supervised semantic ID learning, LMIndexer can generate higher quality semantic ID as identifiers for documents than Ngrams used in SEAL.

Table 11: Hyper-parameter configuration for generative retrieval.

| Parameter | NQ | MACRO-1M |
|---|---|---|
| Optimizer | Adam | Adam |
| Adam $\epsilon$ | 1e-6 | 1e-6 |
| Adam $(\beta_1, \beta_2)$ | (0.9, 0.999) | (0.9, 0.999) |
| Batch size | 2,048 | 2,048 |
| Max steps | 250,000 | 500,000 |
| Max sequence length | 32 | 32 |
| Bean size | 20 | 20 |
| Learning rate | searched in {5e-4, 1e-3, 5e-3} | |
| Backbone LM | T5-base | |

Table 12: Comparison with SEAL (Bevilacqua et al., 2022). Recall@1 and NDCG@5 are used as metrics for NQ320K and Amazon datasets respectively.

| Model | NQ320k | Amazon-Beauty | Amazon-Sports | Amazon-Toys |
|---|---|---|---|---|
| bm25 | 0.2970 | 0.2490 | 0.1898 | 0.2085 |
| Dual Encoder | 0.5360 | 0.2565 | 0.2556 | 0.2805 |
| rq-VAE indexer | 0.6480 | 0.2710 | 0.2606 | 0.2511 |
| HC indexer | 0.6439 | 0.2172 | 0.1979 | 0.2379 |
| SEAL | 0.5698 | 0.1271 | 0.2011 | 0.1035 |
| LMINDEXER | **0.6631** | **0.3187** | **0.2870** | **0.2865** |

## A.6 DUPLICATION STUDY OF SEMANTIC IDS

The duplication issue is very important in learning self-supervised semantic IDs. To alleviate this issue, we propose a contrastive objective in Section 3.2 to promote distinction between documents that previously shared the same prefix and encourage them to obtain different ID for the next position (alleviate duplication). The effectiveness of this design is shown in Figure 8. We can find that during self-supervised learning, if the contrastive objective is added, the difference ratio on next ID position of documents sharing ID prefix is larger and the diversity (perplexity) of IDs on the next position is larger, which means that the duplication issue is alleviated.

We also plot the density curve of the number of documents assigned to each semantic ID after self-supervised learning. The results are shown in Figure 7 We can find that the semantic IDs learned by LMIndexer are quite distinguishable since most IDs contain less than 5 documents. While it is nearly impossible to guarantee zero duplication after self-supervised training since there can be documents that have nearly the same semantics, we simply add another final ID position to distinguish them.

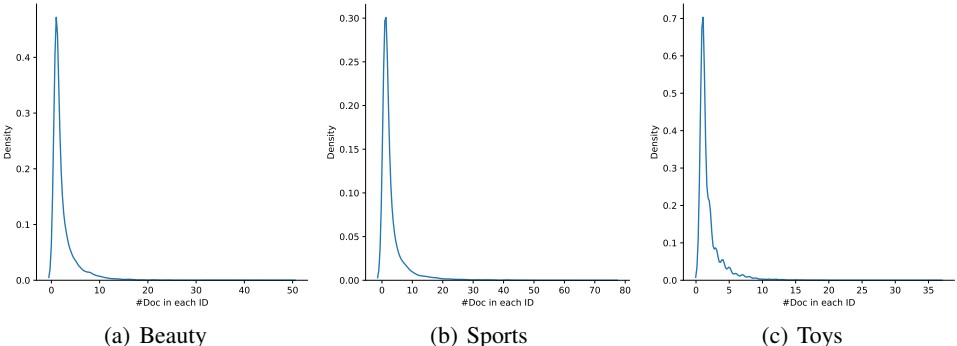

(a) Beauty     (b) Sports     (c) Toys

Figure 7: Study of LMINDEXER's semantic ID duplication problem.

## A.7 MORE QUALITY STUDIES OF SEMANTIC ID

In this section, we conduct a detailed study on the quality of the learned semantic IDs from LMIN-DEXER on Amazon-Beauty dataset. For each product $d$ in the dataset, its learned semantic ID is

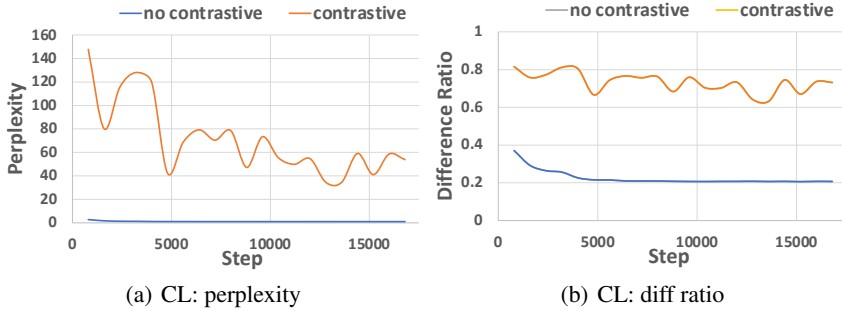

(a) CL: perplexity

(b) CL: diff ratio

Figure 8: Contrastive learning analysis: Documents sharing prefix ID tend to have similar next position ID (low diff ratio) and low diversity (low perplexity) without contrastive objective (blue).

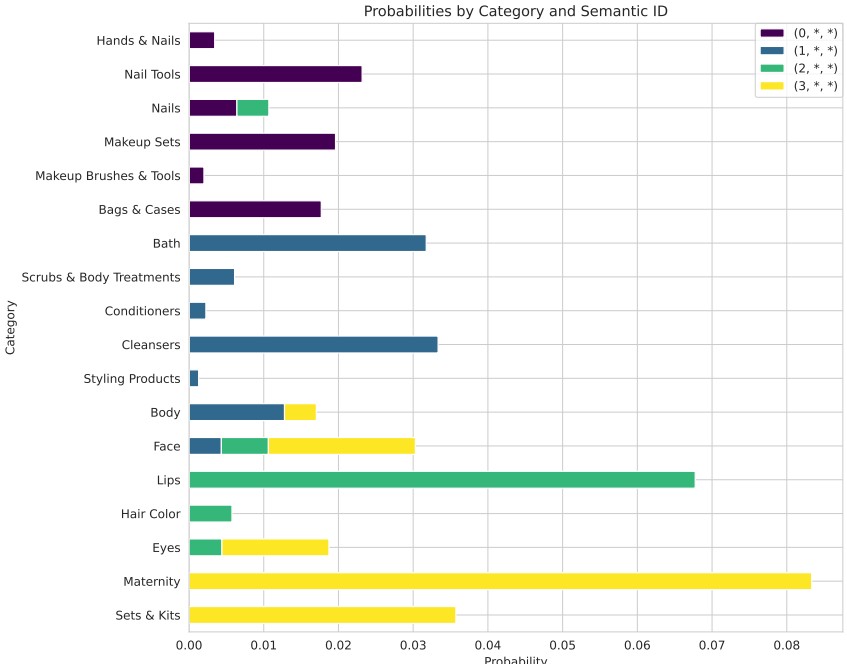

Figure 9: The ground-truth category distribution for all the items in the Amazon-Beauty dataset is colored by the value of the first ID $c^1$.

represented as $c_d = c_d^1 c_d^2 c_d^3$. We randomly select four $c_d^1$ values (*i.e.,* 0, 1, 2, 3) and analyze the products whose $c_d^1 \in \{0, 1, 2, 3\}$. The results are shown in Figure 9 and Figure 10.

In Figure 9, we summarize each item's category using $c^1$ to visualize $c^1$-specific categories in the overall category distribution of the dataset. As shown in Figure 9, $c^1$ captures the coarse-grained category of the item. For example, $c^1 = 1$ contains most of the products related to "Bath". Similarly, majority of items with $c^1 = 0$ are "Tool" and "Make-up" products for nails.

We also visualize the hierarchical structure of LMINDEXER learned Semantic IDs by fixing $c^1$ and visualizing the category distribution for all possible values of $c^2$ in Figure 10. We again found that the second ID $c^2$ further categorizes the coarse-grained semantics captured with $c^1$ into fine-grained categories.

### A.8 CODEBOOK SIZE STUDY

The codebook size is set as a hyperparameter in our model design. We conduct experiments on Amazon-Beatuy dataset to study how codebook size will influence the quality of the learned semantic indexer LMINDEXER. The results are shown in Figure 11. From the result, we can find that the

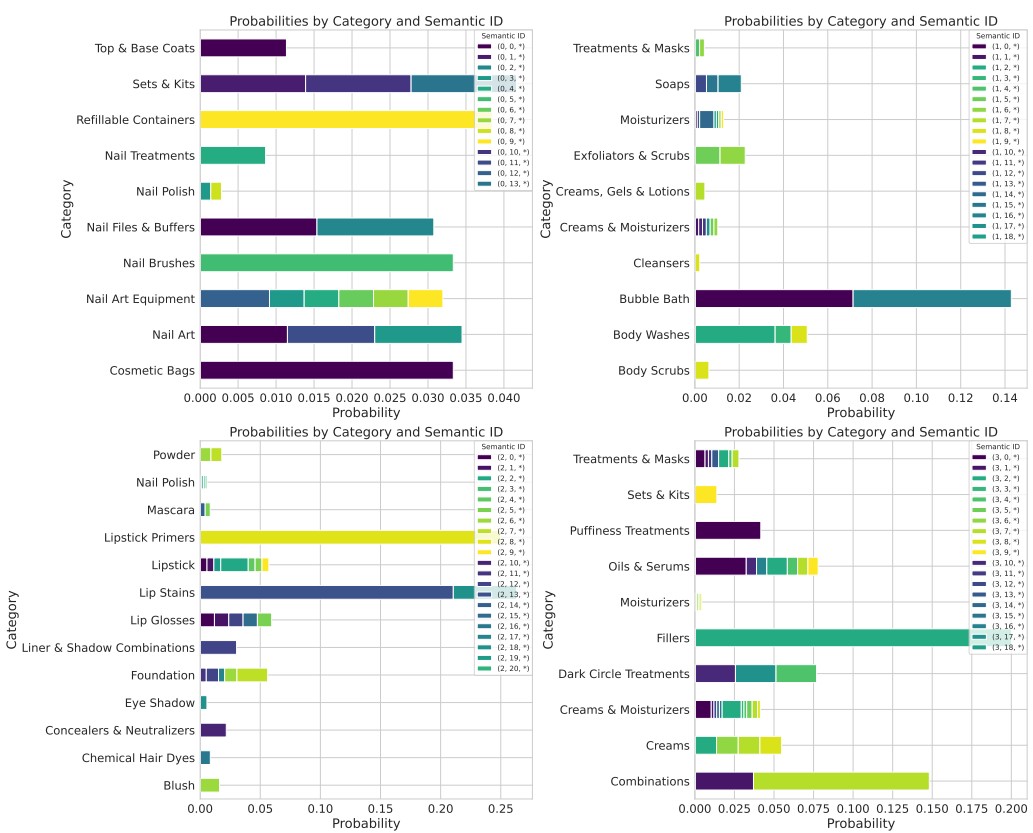

Figure 10: The category distributions for items having the Semantic ID as $(c^1, *, *)$, where $c^1 \in \{0, 1, 2, 3\}$. The categories are colored based on the second semantic token $c^2$.

downstream task performance increases as codebook size increases. It is intuitive, since the larger the codebooks are, the more information they can contain.

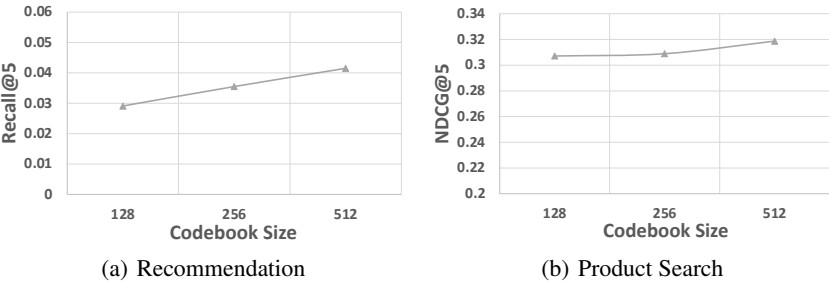

(a) Recommendation                    (b) Product Search

Figure 11: Codebook size study on Amazon-Beauty.

## A.9   DEFINITION OF AMI

The Adjusted Mutual Information (AMI) score (Vinh et al., 2009) is a measure used in statistics and information theory to quantify the agreement between two clusters (in our experiments, the two clusters refer to ground truth category clusters and Semantic ID clusters) while correcting for chance. It is an adjustment of the Mutual Information (MI) score that accounts for the fact that MI is generally higher for clusters with a larger number of clusters, thus providing a normalized score that is more comparable across different clusters.

## A.10   LATENCY ANALYSIS

We conduct latency analysis to compare the time cost of search inference for different methods on Amazon-Beauty dataset. We measure the total latency of product search on the whole Amazon-Beauty test set. The results are shown in Table 13. From the result, the inference latency of our method is comparable with rq-VAE indexer and HC indexer and is much smaller than SEAL.

Table 13: Latency analysis.

| Model | Latency |
|---|---|
| rq-VAE indexer | 13.66s |
| HC indexer | 12.85s |
| SEAL | 21min |
| LMINDEXER | 12.21s |

## A.11   MORE BASELINES FOR SEMANTIC ID QUALITY STUDY (SECTION 4.2)

In this section, we advance the rq-VAE indexer and HC indexer by adopting embeddings from dual-encoder trained on target corpus with contrastive learning Gao et al. (2021) rather than adopting embeddings from off-the-shelf text encoders. In this case, the embeddings are adapted to the target corpus domain and are of better quality. We show the quantitative evaluation of semantic IDs generated by these baselines in Table 14. From the result, we can find that the semantic IDs generated by embeddings obtained after in-domain contrastive learning are better than those generated by embeddings from off-the-shelf text encoders. However, LMIndexer can outperform both baselines, which demonstrates its effectiveness in learning semantic IDs with self-supervision.

## A.12   STUDY OF THE NUMBER OF LAYERS IN THE RECONSTRUCTOR

We conduct a study to explore how the reconstructors of different capabilities can affect the learned semantic indexer in the self-supervised ID learning phase. We try reconstructors with 2 layers and 3 layers in Amazon-beauty dataset and the results shown in Table 15. From the result, we can find that as the reconstructor layer increases (the reconstructor becomes more powerful), the quality of

Table 14: More baselines on ID quantitative study (AMI).

| Model | Beauty | Sports | Toys |
|---|---|---|---|
| rq-VAE indexer (BERT) | 0.2654 | 0.2774 | 0.3154 |
| HC indexer (BERT) | 0.2428 | 0.2387 | 0.2729 |
| rq-VAE indexer (In-domain Contrastive) | 0.3100 | 0.2695 | 0.3126 |
| HC indexer (In-domain Contrastive) | 0.2771 | 0.2622 | 0.2968 |
| LMINDEXER | **0.3563** | **0.4163** | **0.3536** |

the semantic indexer and its generated semantic IDs decreases. This is because more knowledge is learned inside the reconstructor rather than in the semantic indexer during self-supervised learning.

Table 15: Study of the number of layers in reconstructor on Amazon-Beauty dataset. AMI, Recall@5, and NDCG@5 are used as metrics for ID quality study, recommendation, and retrieval.

| Model | ID quality | Recommendation | Retrieval |
|---|---|---|---|
| LMINDEXER (Recon 1 layer) | **0.3563** | **0.0415** | **0.3187** |
| Recon 2 layers | 0.2390 | 0.0284 | 0.2528 |
| Recon 3 layers | 0.1679 | 0.0281 | 0.2522 |

## A.13 HUMAN EVALUATION OF SEMANTIC ID QUALITY

In this section, we conduct a human evaluation of the learned semantic IDs from different methods. We adopt a three-step pipeline to conduct the evaluation: 1) We Randomly select product pairs in the Amazon-sports dataset that share the first two IDs $c^{<2} = c^1 c^2$ (20 pairs for each method). 2) We ask four trained annotators to evaluate if the two products in each pair are semantically related to each other. 3) We finally calculate the accuracy of each method. The results are shown in Figure 16. From the result, our LMIndexer can outperform baseline methods by a large margin.

Table 16: Human Evaluation of Semantic ID quality.

| Model | Accuracy |
|---|---|
| rq-VAE indexer | 0.7375 |
| HC indexer | 0.5375 |
| LMINDEXER | 0.7750 |

