# OpenReview forum: "Language Models as Semantic Indexers"
_ICLR.cc/2024/Conference — Submitted to ICLR 2024_

### Official Review · Reviewer_2Nr6 · 2023-10-30

**Soundness:** 2 fair
**Presentation:** 3 good
**Contribution:** 3 good
**Rating:** 6
**Confidence:** 4

**Summary:**

Semantic id is an interesting and promising topic.
This task reform counting id to the identifier with semantic, which will faciliate many intelligence application including e-commerce.
This work presents a self-supervised framework to learn semantic IDs with a generative language model.
By progressive learning and contrastive learning, the authors achieve sequential discrete semantic indexier.
The paper is well-motivated and well-written.

Major Concerns:
1. We have not seen a full section to anlayze the semantic IDs by demonstrating all sorts of key examples.
As a rising domain, many readers may wonder what the semantic IDs exactly look like and how they benefit the downstream tasks.
2. In your experiments, could you provide some human annotator evaluation for the semantic of your generated IDs?
This is a key experiment, though we have seen enough assessment.

**Strengths:**

Semantic id is an interesting and promising topic.
This task reform counting id to the identifier with semantic, which will faciliate many intelligence application including e-commerce.
This work presents a self-supervised framework to learn semantic IDs with a generative language model.
By progressive learning and contrastive learning, the authors achieve sequential discrete semantic indexier.
The paper is well-motivated and well-written.

**Weaknesses:**

Experimental analysis is not sufficient to support the contribution.

**Questions:**

Major Concerns:
1. We have not seen a full section to anlayze the semantic IDs by demonstrating all sorts of key examples.
As a rising domain, many readers may wonder what the semantic IDs exactly look like and how they benefit the downstream tasks.
2. In your experiments, could you provide some human annotator evaluation for the semantic of your generated IDs?
This is a key experiment, though we have seen enough assessment.

---

> ### Author Response · Authors · 2023-11-17
> **Response to Reviewer 2Nr6**
>
> Thank you so much for your thoughtful review!
>
> Regarding your questions:
> 1. **What do semantic IDs look like?** We thank the reviewer for raising this comment. Accordingly, we add a semantic ID study section in Appendix A.7 to show what the IDs look like and what documents are assigned to those IDs. From the result, we can find that documents/products sharing similar IDs are semantically similar to each other. In addition, the first c^1 captures coarse-grained functions (e.g., Bath products) while the second c^2 will dig into more fine-grained semantics (e.g., Soaps, Shampoo).
>
> 2. **Human evaluation.** We agree with the reviewer that a human evaluation can make the experiment on ID quality more comprehensive. We add a human evaluation by 1) Randomly selecting product pairs (20 pairs for each method) in the Amazon-sports dataset which share the first two IDs c^{<2}=c^1c^2. 2) Ask four trained annotators to evaluate if the two products in each pair are semantically related to each other. 3) Calculate the accuracy of each method. The results are shown below:
>
> | Model              | Accuracy |
> |--------------------|--------|
> | rq-VAE indexer     |  0.7375 |
> | HC Indexer         |  0.5375 |
> | Ours               | 0.7750 |
>
> From the result, our LMIndexer can outperform baseline methods by large margins. The human evaluation form can be found at https://docs.google.com/forms/d/e/1FAIpQLSfwtA4C6IY-YF4soWBWOqKMtJOso33p7DdUJic1_tE8utv5ww/viewform.

---

> ### Author Response · Authors · 2023-11-23
> **Kind Reminder**
>
> We wish to express our sincere gratitude once again to the reviewers for their valuable contributions and considerate feedback. We would like to gently bring to the reviewers' attention that the interaction phase between authors and reviewers is nearing completion (within 12 hours).
>
> Given the inclusion of the new experiments, we kindly inquire whether the reviewers might consider assigning a more favorable evaluation to our submission. Should you have any further insights to share, we are more than willing to sustain our discussion until the deadline.

---

### Official Review · Reviewer_HnqK · 2023-10-31

**Soundness:** 1 poor
**Presentation:** 3 good
**Contribution:** 2 fair
**Rating:** 5
**Confidence:** 4

**Summary:**

Discrete semantic IDs are useful in information retrieval tasks. They are often learned by performing some sort of hierarchical clustering over off-the-shelf item representations which is not ideal as they may not be aligned with the downstream task. This paper presents an approach to learn discrete semantic IDs of items in a self-supervised manner. The proposed approach uses a transformer decoder architecture which encodes the item description and decodes it into semantic IDs which is further coupled with a small transformer model that consumes these semantic IDs, item description and tries to perform MLM task. The paper also describes and suggests ways to circumvent the challenges encountered while learning these semantic IDs.

**Strengths:**

- The paper is in general well-written and easy to follow
- The approach is novel for learning semantic IDs in information retrieval, and the optimization challenges in such learning problems is highlighted

**Weaknesses:**

My major concerns with the paper are the weak evaluation and baselines, and overall the training seems to need a lot of bells and whistles to succeed.
- DPR dual encoder is not a strong baseline; DPR is almost 10% behind SOTA dual-encoder approaches on standard benchmarks
- Baselines in section 4.2 are weak since they are using an off-the-shelf text encoder and hence have no knowledge about the task; a very simple baseline that could be tried here is to train a dual-encoder model on this corpus and then cluster the embeddings

**Questions:**

- The learned semantic ID lengths seem to be very small (1-3), is this because of training instability when scaling to larger ID lengths? Do the other generative baselines also use the same ID lengths?
- Why not use the full MS-Marco dataset instead of the 1M sampled one?
- How does the performance gets affected when using a more powerful reconstructor? perhaps an ablation on the number of layers in the reconstructor might be helpful here
- Is the contrastive loss $\mathcal{L}_{\text{contrastive}}$ computed over all documents?

---

> ### Author Response · Authors · 2023-11-17
> **Response to Reviewer HnqK**
>
> Thank you so much for your thoughtful review!
>
> Regarding your questions:
>
> 1. **Simple baseline in 4.2.** We thank the reviewer for proposing this baseline. According to your comment, we add experiments on Amazon datasets by first training a dual-encoder on the target corpus with contrastive learning [1] and adopting hierarchical clustering or rq-VAE to obtain semantic IDs based on the learned embeddings, denoted as rq-vae-contrastive and HC-contrastive, respectively. The results for ID quantitative study are shown below (can also be found in Appendix A.11):
>
> | Model              | Beauty | Sports | Toys   |
> |--------------------|--------|--------|--------|
> | rq-VAE-bert        | 0.2654 | 0.2774 | 0.3154 |
> | HC-bert            | 0.2428 | 0.2387 | 0.2729 |
> | rq-VAE-contrastive | 0.3100 | 0.2695 | 0.3126 |
> | HC-contrastive     | 0.2771 | 0.2622 | 0.2968 |
> | Ours               | 0.3563 | 0.4163 | 0.3536 |
>
> From the result, we can find that the semantic IDs generated by embeddings obtained after in-domain contrastive learning are better than those generated by embeddings from off-the-shelf text encoders. However, LMIndexer can outperform both baselines, which demonstrates its effectiveness in learning semantic IDs with self-supervision.
>
> 2. **Ablation study on reconstructor.** We agree with the reviewer that adding such a study can make the experiments more comprehensive. We try reconstructors with 2 layers and 3 layers in Amazon-beauty dataset and the results are shown below (can also be found in Appendix A.12):
>
> | Model              | ID AMI |   Recommendation (R@5)  | Product Search (NDCG@5)   |
> |--------------------|--------|--------|--------|
> | Ours (recon-1)     | 0.3563 | 0.0415 | 0.3187 |
> | recon-2            | 0.2390 | 0.0284 | 0.2528 |
> | recon-3            | 0.1679 | 0.0281 | 0.2522 |
>
> From the result, we can find that as the reconstructor layer increases (the reconstructor becomes more powerful), the quality of the semantic indexer and its generated semantic IDs decreases. This is because more knowledge is learned inside the reconstructor rather than in the semantic indexer during self-supervised learning, which results in ID quality degeneration.
>
>
> 3. **Dual encoder (DPR) baseline.** Thanks for your comments! We would like to argue this point from two perspectives: 1) *Focus*: The focus of our work is to explore a self-supervised learning method to learn semantic IDs for documents and study if the learned semantic indexer and semantic IDs can be adapted to various downstream tasks. We are happy to compare with more baseline methods but we would like to mention that our goal is not to design a model to compete on one specific task. 2) *DPR setting*: In our experiments for DPR, we are adopting the T5-base as the pretrained checkpoint rather than BERT-base in the original DPR setting (code can be found in https://anonymous.4open.science/r/ICLR24-submit-B2E7). This dual encoder setting is demonstrated to be much stronger [2].
>
> 4. **Semantic ID length.**  We illustrate from two perspectives why we set T to be 3. 1) *Performance*. As shown in Figure 5,  as T increases, the performance increase speed becomes smaller. This means that the gain of making T larger is smaller. Since a larger T will cost more time in training and inference, we adopt T=3 in our experiments. 2) *ID duplication*. As shown in Figure 7 in the Appendix, when we set T=3, the duplication issues of IDs are largely alleviated. It means that T=3 can make the ID for different documents distinguishable.
>
> 5. **MS-MACRO dataset.** Thanks for your question. We sample a 1M MS-Macro document subset to evaluate our method following [3].
>
> 6. **Contrastive loss.** The contrastive loss is designed to promote distinction on c^t between documents that previously shared the same c^{<t}, enabling the model to discern finer-grained hierarchical relationships between documents and alleviate ID duplication issues. The contrastive loss is computed over all documents. For each document, we serve other documents that share c^{<t} with it to be negative samples when learning c^t.
>
>
> [1] Gao, et al. Simple Contrastive Learning of Sentence Embeddings. EMNLP 2021.
>
> [2] Ni, et al. Large Dual Encoders Are Generalizable Retrievers. EMNLP 2022.
>
> [3] Pradeep, et al. How Does Generative Retrieval Scale to Millions of Passages?

---

> > ### Comment · Reviewer_HnqK · 2023-11-22
> >
> > Thanks for the response, it cleared some of my doubts but I am still not convinced about the effectiveness of the proposed approach. I think it's an interesting approach but there is not enough evidence to support its usefulness on downstream tasks. I am updating my score to 5 in light of the author's further clarifications.

---

> > > ### Author Response · Authors · 2023-11-23
> > >
> > > Dear Reviewer,
> > >
> > > We appreciate very much your acknowledgment of our response, considering the new experiments, and for raising your score!
> > >
> > > We are glad that we have addressed many of your concerns. Regarding the experiments, we demonstrate the effectiveness of LMIndexer on downstream tasks (including recommendation, product search, and document retrieval) with a notable improvement over competitive baseline methods. Your acknowledgment through an increased score is greatly appreciated, and we remain dedicated to continuous enhancements.
> > >
> > > Best regards,
> > > The authors

---

### Official Review · Reviewer_dU5G · 2023-11-01

**Soundness:** 2 fair
**Presentation:** 3 good
**Contribution:** 3 good
**Rating:** 6
**Confidence:** 3

**Summary:**

The authors propose LMIndexer, a self-supervised method to learn the semantic IDs of documents using sequence-to-sequence models. A semantic ID of a document is a sequence of integers that are indexes/row numbers of a codebook embedding matrix. Three loss functions are designed and used to train the sequence-to-sequence model: a reconstruction loss, a contrastive loss, and a commitment loss. The proposed method is evaluated on three downstream tasks: sequential recommendation, product search, and document retrieval. The results show good improvement over some SOTA methods.

**Strengths:**

-	The proposed method formulates the semantic ID learning problem as a sequence-to-sequence learning method, which is novel according to related work discussed in the paper.
-	Technical challenges are described clearly.
-	SOTA techniques are used in the proposed framework.
-	The experimental results show that the proposed method outperforms some SOTA methods in the three downstream tasks.

**Weaknesses:**

-	The paper says that the proposed method "learns the document’s discrete semantic embeddings and its hierarchical structure simultaneously". But it is not clear what the authors mean by the hierarchical structure of a document, how the proposed method is guaranteed to learn such a structure, and whether the proposed method actually learns such a structure.

-	The size of the semantic ID (T) is set to less than or equal to 3 in the experiments, which is surprisingly small. Figure 5 shows the performance increases with T. Why not trying bigger T values? Would an ID with length 3 be informative enough?

-   How should the codebook size be determined? It seems that only three sizes are tried in the experiments. How does the size of the codebook affect the performance?

-  The performance metric used in ID quantitative study, AMI (in Table 1), is not defined or explained.

-   Not clear what the word clouds picture (Figure 3) is trying to show? The text explaining it is confusing. What do you mean by “two semantic ID prefixes”? Are the two prefixes from the same generated ID or two different IDs?

-   It would be better if authors provided other performance metrics such as latency for the retrieval in comparison with baselines.

-  The authors mentioned the model could be fine-tuned on downstream tasks such as retrieval or recommendation tasks. Fine-tuning would change the model weights and then the previously generated semantic IDs would be changed as well, which may affect the ground-truth ID used in fine-tuning. Does it cause any problem?

**Questions:**

Please see the questions in the above section.

---

> ### Author Response · Authors · 2023-11-17
> **Response to Reviewer dU5G**
>
> Thank you so much for your thoughtful review!
>
> Regarding your questions:
> 1. **Hierarchical structure.** 1) *Meaning*: Learning the hierarchical structure of documents refers to understanding documents from different semantic granularity and reflecting them in their semantic IDs (The preliminary set of IDs should predominantly encapsulate coarse-grained semantics, with successive IDs delving deeper into nuanced specifics). A more detailed illustration is added in Appendix A.7. 2) *Techniques*: The LMIndexer is proposed to learn the semantics into the IDs with self-supervised sequential discrete auto-reconstruction and capture the different granularity of semantics into their IDs with progressive training and contrastive objective. 3) *Results*: We show the learned hierarchy in Figure 9 and Figure 10 in the Appendix, where we can find that the products sharing the same first ID have a similar coarse-grained function (e.g., Bath products), while products sharing the first and second ID have related fine-grained functions (e.g., Soaps, Shampoo).
>
> 2. **Size of the semantic ID (T).** We illustrate from two perspectives why we set T to be 3. 1) Performance. As shown in Figure 5, the performance increases speed as T increase becomes smaller. This means that the gain of making T larger is smaller. Since a larger T will cost more time in training and inference, we adopt T=3 in our experiments. 2) ID duplication. As shown in Figure 7 in the Appendix, when we set T=3, the duplication issues of IDs are largely alleviated. It means that T=3 can make the ID for different documents distinguishable.
>
> 3. **Codebook size.** The codebook size is set as a hyperparameter in our model design. We agree with the reviewer that showing the study of codebook size can make our research more comprehensive. The results are shown below (added in Appendix A.8):
>
> | Model              |   Recommendation (R@5)  |   Product Search (NDCG@5)   |
> |--------------------|--------|--------|
> | 128            | 0.0291 | 0.3071 |
> | 256            | 0.0355 | 0.3089 |
> | Ours (512)     | 0.0415 | 0.3187 |
>
> From the result, we can find that the downstream task performance generally increases as codebook size increases. It is intuitive since the larger the codebooks are (before overfitting), the more information they can represent.
>
> 4. **Definition of AMI.** The Adjusted Mutual Information (AMI) score is a measure used in statistics and information theory to quantify the agreement between two clusters (in our experiments, the two clusters refer to ground truth category clusters and Semantic ID clusters) while correcting for chance. It is an adjustment of the Mutual Information (MI) score that accounts for the fact that MI is generally higher for clusters with a larger number of clusters, thus providing a normalized score that is more comparable across different clusters. We have added the definition of AMI in Appendix A.9 according to the reviewer's suggestion.
>
> 5. **Word clouds.** The word clouds are shown for the semantic ID qualitative study. “Two semantic ID prefixes” refers to two different c^{<=2}=c^1c^2 (e.g., ID (0,0,*) and ID (1,2,*)). We try to print out the word clouds to summarize what documents are sharing IDs (0,0,*) and what documents are sharing IDs (1,2,*) respectively. We can find that the two groups of documents are related to beach/sand set toys (corresponding to ID (0,0,*)), and star war toys (corresponding to ID (1,2,*)) respectively. This demonstrates that the IDs acquired through LMIndexer convey a higher degree of semantic relevance (enabling similar documents to have similar semantic IDs). We have added a more comprehensive ID qualitative study section in Appendix A.7.
>
> 6. **Latency analysis.** We conduct latency analysis on baseline methods and LMIndex on Amazon-beauty dataset. We measure the total latency of product search on the whole Amazon-beatuy test set. The results are shown below (details can be found in Appendix A.10):
>
> | Model              | Beauty |
> |--------------------|--------|
> | rq-VAE indexer     |  13.66s |
> | HC indexer         |  12.85s |
> | SEAL               |  21min |
> | Ours               | 12.21s |
>
> From the result, the inference latency of our method is comparable with the rq-VAE indexer and HC indexer and is much smaller than SEAL.
>
> 7. **Semantic ID & Fine-tuning.** This is a great point. After the self-supervised semantic ID learning, we have a semantic indexer that can capture document semantics into their IDs. The semantic IDs for all documents will be generated with this semantic indexer and will not be updated in downstream tasks. During downstream task fine-tuning, only the parameters inside the LMIndexer are updated (semantic IDs for documents are fixed), where the model tries to learn the mapping between new types of input text (e.g., user history in recommendation and query in retrieval) to fixed document semantic IDs.

---

> ### Author Response · Authors · 2023-11-23
> **Kind reminder**
>
> We wish to express our sincere gratitude once again to the reviewers for their valuable contributions and considerate feedback. We would like to gently bring to the reviewers' attention that the interaction phase between authors and reviewers is nearing completion (within 12 hours).
>
> Given the inclusion of the new experiments and further explanation of our methods, we kindly inquire whether the reviewers might consider assigning a more favorable evaluation to our submission. Should you have any further insights to share, we are more than willing to sustain our discussion until the deadline.

---

### Official Review · Reviewer_4DWK · 2023-11-01

**Soundness:** 3 good
**Presentation:** 3 good
**Contribution:** 2 fair
**Rating:** 5
**Confidence:** 4

**Summary:**

This paper formulates the problem of learning semantic IDs by simultaneously capturing the document's semantic representations and its hierarchical structure. It introduces an innovative self-supervised approach designed to acquire semantic IDs directly from the input document using a generative language model. Experimental results on five datasets from various domains demonstrate that the proposed method consistently outperforms competitive baselines by a significant margin.

**Strengths:**

1. This paper presents the "LMINDEXER" approach as a solution to the challenges inherent in generating semantic IDs from textual data. The approach is carefully crafted to capture both the semantic representations and hierarchical structure of documents simultaneously.

2. The paper demonstrates the effectiveness of the LMINDEXER approach through empirical evidence gathered from experiments on three distinct downstream tasks, utilizing data from diverse domains.

3. The paper exhibits a well-organized structure and offers an easily digestible reading experience.

**Weaknesses:**

1. While this paper presents an approach termed LMINDEXER, it's important to note that the novelty of the method is somewhat limited. Additionally, the paper lacks a comprehensive discussion of related work, including notable prior efforts that have explored the use of encoders for text encoding and decoders for reconstruction in the context of information retrieval. Several works, such as [1], [2], and [3], have examined similar techniques and deserve acknowledgment for their contributions to the field.

[1] Less is More: Pretrain a Strong Siamese Encoder for Dense Text Retrieval Using a Weak Decoder (EMNLP 2021)

[2] A Contrastive Pre-training Approach to Learn Discriminative Autoencoder for Dense Retrieval (CIKM 2022)

[3] RetroMAE: Pre-Training Retrieval-oriented Language Models Via Masked Auto-Encoder (EMNLP 2022)


2. In the context of document retrieval, it would be beneficial to broaden the comparison to include more generative retrieval baselines. For instance, evaluating how the LMINDEXER approach compares to FM-index-based SEAL or other recent generative retrieval methods would provide a more comprehensive understanding of its performance. Focusing solely on comparisons with DSI, which may be considered a relatively weaker baseline, might not offer a complete picture of the method's capabilities.

3. While the proposed self-supervised approach has the potential to address the "new document problem" by automatically acquiring semantic IDs, it would be valuable to see experimental results that explicitly address this issue. Incorporating experiments that involve adding new documents to the model and assessing its adaptability and performance in such scenarios would provide a more robust evaluation.

4. An important consideration when using automatically generated semantic IDs is the possibility of duplication. It is crucial to include a discussion or explanation of how the LMINDEXER approach handles or mitigates this potential issue. A detailed exploration of the approach's robustness in preventing or addressing duplication would enhance the paper's completeness and practicality.

**Questions:**

1. The authors should consider expanding their related work section to include the most recent developments in generative Information Retrieval (IR) techniques. Additionally, they should include a comparative analysis of these recent works alongside the proposed LMINDEXER method in the experimental section. This would provide a more comprehensive overview of how LMINDEXER stacks up against the state-of-the-art in generative IR.

2. It is essential for the authors to address the issue of handling new documents within the LMIndexer framework. The paper should discuss how this framework manages the addition of new documents, what mechanisms or strategies are employed, and the performance of the LMIndexer approach in comparison to baseline methods when confronted with this "new document" scenario. This analysis would help assess the adaptability and robustness of the approach.

3. To ensure that distinct semantic IDs are assigned to different documents, the paper should provide detailed explanations and discussions regarding the mechanisms and safeguards in place within the LMIndexer framework. Experimental results or case studies showcasing how the system maintains distinct semantic IDs for various documents would add substantial value to the paper, reinforcing its practicality and effectiveness in preventing semantic ID duplication.

---

> ### Author Response · Authors · 2023-11-17
> **Response to Reviewer 4DWK**
>
> Thank you so much for your thoughtful review!
>
> Regarding your questions:
> 1. **More related works.** We thank the reviewer for pointing out the related papers and we are happy to add them to the related work section (added to the “Self-supervised Learning with Language Models” section). We would like to emphasize the novelty of LMIndexer compared with these works: 1) *Focus*: LMIndex mainly focuses on learning semantic IDs via self-supervised learning, which is different from existing works’ focus on pretraining a text encoder for dense retrieval. 2) *Techniques*: Since semantic IDs lie in a discrete and sequential fashion which is different from a soft embedding for dense retrieval, we propose a new sequential discrete auto-reconstruction pipeline and introduce solutions for challenges including Reconstructor collapse, Posterior collapse, and ID diversity. 3) *Task*: We explore LMIndex on product recommendation, product retrieval, and document retrieval and existing works mainly focus on document retrieval.
>
> 2. **Comparison with SEAL.** We thank the reviewer for pointing out the baseline. We add their results on Amazon product search datasets and NQ320k (code to reproduce is uploaded to https://anonymous.4open.science/r/ICLR24-submit-B2E7/):
>
> | Model              | Beauty | Sports | Toys   |  NQ320k |
> |--------------------|--------|--------|--------|--------|
> | rq-VAE indexer        |  0.2710 |  0.2606 | 0.2511 |  0.6480  |
> | HC indexer     |  0.2172 | 0.1979 |  0.2379 |   0.6439 |
> | SEAL               | 0.1271 | 0.2011 | 0.1035 |   0.5698 |
> | Ours               | 0.3187 |  0.2870 | 0.2865 |   0.6631|
>
> We also add the experimental results in Appendix A.5 in our paper. From the result, we can find that our method outperforms SEAL significantly. The main reason is that after the self-supervised semantic ID learning, LMIndexer can generate higher quality semantic ID as identifiers for documents than Ngrams used in SEAL.
>
> 3. **New document problem.** We conduct experiments to study this problem in Section 5.2 (Table 4). After the semantic indexer is finetuned on the downstream product search task, we add 100 new documents into the corpus. We use the semantic indexers before fine-tuning on the downstream task (including rq-VAE indexer, HC indexer, and LMIndexer) to give semantic IDs for new documents and test how the fine-tuned semantic indexers (on the downstream task) can generalize to the new documents. The results are shown in Table 4, where LMIndexer outperforms the baselines significantly. The reason is that LMIndexer is trained to achieve semantic IDs on a large corpus with self-supervision. This enables the LMIndexer to learn the projection between text semantics and semantic IDs, which can be generalized to new documents and applied to downstream tasks. We will make this part more clear in our revision.
>
> 4. **ID Duplication.** This is a great point. We add a section to discuss this issue in Appendix A.6. We agree with the reviewer that the duplication issue is very important here. To alleviate this issue, we propose a contrastive objective in Section 3.2 to promote distinction between documents that previously shared the same learned ID and encourage them to obtain different IDs for the next position (alleviate duplication). The effectiveness of this design is shown in Figure 8 in Appendix. We can find that during self-supervised learning, if the contrastive objective is added, the difference ratio on the next ID position of documents sharing learned ID is bigger and the diversity (perplexity) of IDs on the next position is larger, which means that the duplication issue is alleviated. We also show in Figure 7 in the Appendix that the semantic IDs learned by LMIndexer are quite distinguishable. While it is nearly impossible to guarantee zero duplication since there can be documents that have very similar semantics, we simply add another final ID position to distinguish them.

---

> > ### Comment · Reviewer_4DWK · 2023-11-21
> >
> > Thanks to the authors and the very detailed reply.
> >
> > As the extended experiments on new-document and id-duplicate problems, I can confirm that these two problems exist in the proposed method. For example, the performance of the new-document setting is only 0.0455 on Recall@50 compared to the original 0.3187 on NDCG@5. Furthermore, the duplicate ID problem is significant, because 20% or more documents have the same IDs. I am not sure because the Figure 7 is over-smoothed. Finally, I think the performance of SEAL needs to be verified because I think it at least outperforms the DSI model.

---

> > > ### Author Response · Authors · 2023-11-21
> > > **Response to Reviewer 4DWK**
> > >
> > > Thank you so much for your feedback!
> > >
> > > Regarding your comments:
> > > 1. **New Document.** 1) *Comparison with baselines*: We compare the performance of our method with baselines on the new document setting in Table 4, where LMIndexer outperforms baselines significantly. This demonstrates the advancement of LMIndexer for zero-shot retrieval compared with semantic ID baselines. 2) *Generalizing to new documents is not our focus*. We want to argue that zero-shot retrieval itself is a hard research problem [1]. We agree with the reviewer that it can be an interesting future direction for semantic IDs, but it is not the focus of this paper (this paper is mainly proposed for learning self-supervised semantic IDs).
> > > 2. **ID Duplication.** 1) *Depending on corpus*. We would like to argue that the ID duplication problem is not the problem of our method, but originated from the nature of the corpus. In text corpus, there often exists documents that are nearly the same as each other, or just different in several words. For example, in the Amazon-toys dataset (which we show the duplication results for), there are items “Monster High Skull Shores Ghoulia”, “Monster High Skull Shores Abbey” and “​​Monster High Skull Shores Lagoona”. In these cases, it is extremely difficult to assign IDs only based on text semantics. 2) *Solution*. We solve this problem by simply adding a final ID position which is not learned based on semantics to distinguish the products sharing the same IDs. By doing this, the IDs will not only contain the text semantics of the original document/products but are also 100% distinguishable.
> > > 3. **SEAL.** 1) *Not exactly DSI*. We would like to argue that the baselines are not exactly DSI, but are enhanced with pseudo queries during fine-tuning following [2][3]. It is also demonstrated in [2] that this setting can contribute to better performance compared with SEAL. 2) *Careful Experiments with SEAL*: We carefully follow the instructions and settings (hyperparameters, early stops, etc) in the SEAL code to preprocess the dataset and train the SEAL model. The code to reproduce can be found at https://anonymous.4open.science/r/ICLR24-submit-B2E7/.
> > >
> > >
> > > We thank the reviewer for the insightful comments and will try our best to make the illustration as clear as possible. We are grateful for your time helping us further improve the paper and would appreciate it if you could reconsider the score.

---

### Author Response · Authors · 2023-11-17
**General Response**

Dear Reviewers,

We sincerely appreciate your valuable feedback and suggestions. Our work has been revised based on your reviews. We highlighted changes in the manuscript using blue color.

We also want to thank the Reviewers for noting the strengths of our paper, namely:

- The problem addressed in our paper is important and promising. (2Nr6)
- Our proposed method is novel, substantial, and sound. (dU5G, HnqK)
- The paper is clearly written. (4DWK, HnqK, 2Nr6)
- The empirical results are consistent, solid, and convincing. (4DWK, dU5G)

We have addressed the individual questions of reviewers in separate responses. In the revised version, we incorporated all reviewers' suggestions by adding more clarification of our method, more experimental results and baselines, as well as deeper model analysis. Here we briefly outline the updates to the revised submission for the reference of reviewers.

- [Section 2: Related Works] We add more related works which are pointed out by reviewers. (4DWK)
- [Appendix A.5] We add a comparison between our methods with SEAL. (4DWK)
- [Appendix A.6] A Duplication study of semantic IDs is performed and the effectiveness of our design to alleviate the duplication is studied. (4DWK, dU5G, HnqK)
- [Appendix A.7] More quality studies of semantic ID are conducted to understand what the semantic ID looks like and what the learned ID hierarchy is. (dU5G, 2Nr6)
- [Appendix A.8] We conduct a study on how the codebook size will affect the quality of LMIndexer on downstream tasks. (dU5G)
- [Appendix A.9] We add the definition of AMI metric. (dU5G)
- [Appendix A.10] Latency analysis is conducted to compare LMIndexer with baselines. (dU5G)
- [Appendix A.11] We add semantic IDs generated by embeddings inferred by dual encoder which is trained via contrastive learning on the target corpus. (HnqK)
- [Appendix A.12] The study of the number of layers in the reconstructor is performed. (HnqK)
- [Appendix A.13] We conduct human evaluation to measure the quality of the semantic IDs generated by our method and baselines. (2Nr6)

In closing, we thank the Reviewers again for their time and valuable feedback. If there are further concerns, please let us know, and we will be happy to address them.

---

### Author Response · Authors · 2023-11-22
**Kind Reminder**

Dear reviewers,

We want to sincerely thank you again for your involvement and thoughtful feedback! We hope our response addresses your questions. We would like to gently remind you that we are approaching the end of the discussion with author involvement. In light of the newly added baselines (reviewer 4DWK, HnqK), further LMIndexer studies (reviewer 4DWK, dU5G, HnqK, 2Nr6), and model clarifications (reviewer dU5G, HnqK), we are kindly asking if you are willing to give a higher assessment to our submission. If you have any more thoughts, we are happy to continue our discussion until the deadline.

Thank you so much!

---

### Meta-Review · Area_Chair_QZ77 · 2023-12-23

**Metareview:**

The paper proposes a method, LMIndexer, to learn discrete representations of documents. The method is based on a modified quantized auto-encoder. The encoder is a full Transformer. The decoder is one layer Transformer decoder. The training includes a reconstruction loss, a contrastive loss, and a commitment loss. The proposed method is evaluated on three downstream tasks: sequential recommendation, product search, and document retrieval. The results show good improvement over some SOTA methods.

Strengths:
1. The method is novel in its design of the training loss.
2. The experiments on multiple datasets show the end-task performance of the proposed method.

Weakness:
1. The paper does not include ablation analysis. It is hard to identify which of the three training loss, or the downstream fine-tuning is the source of gain.
2. One missing important baseline is the multi-index product quantization based on VAE (or simple Roberta embeddings). The commonly used approach is multi-index based on hamming distance. Again, this is a strong baseline and can be easily applied to new documents.

During the discussion, the authors provided additional experimental results regarding comparison with SEAL, latency, and human evaluation.

**Justification For Why Not Higher Score:**

Missing ablation to identify the contribution of the components. Missing baseline.

**Justification For Why Not Lower Score:**

N/A

---

### Decision · Program_Chairs · 2024-01-16

Reject